# Modelling of wind flows over realistic forests with LES

Hugo Olivares-Espinosa and Johan Arnqvist Department of Earth Sciences, Uppsala University, Visby, Sweden **Correspondence:** Hugo Olivares-Espinosa (hugo.olivares@geo.uu.se)

## Abstract.

An LES based model for the simulation of wind flows over realistic forests and topography is presented. Terrain elevation as well as forest density maps from airborne laser scans are employed to investigate the importance of specific model choices related to capturing upstream terrain effects on the wind resource. The study is divided in three parts. Firstly, an extended

- verification process over idealized conditions is carried out. Secondly, a validation where the model is compared to field measurements acquired in the south-east of Sweden and finally an assessment of the forest and terrain footprint based on variations of the surface representation. The results show an agreement of turbulence statistics compared to the literature when forest is explicitly modelled, following expected trends as a function of the tree density. When the forest is explicitly modelled the impact of the ground roughness becomes insignificant, even for an unrealistically sparse forest. The study also demonstrates that
- a model relying only on ground roughness yields notable differences in the turbulence characteristics. This is partly attributed to the inability of the model to reproduce sufficient drag for forest-equivalent values of roughness length  $z_0$  while maintaining the applicability of wall functions, which can impose strict limitations on the grid near ground. This is further complicated by the problem of converting realistic, heterogeneous forests fields to  $z_0$ . Moreover, turbulence statistics in the roughness sublayer are affected by the lack of vertical permeability. The validation shows that the model is able to capture the flow characteristics
- imprinted by different surface features on the wind along three distinctive wind directions. Vertically separated spectral coherence from the LES is slightly below compared to the IEC standard, which can be attributed to the reference velocities used in the normalization of the frequency. The footprint study shows that the heterogeneity of a realistic forest produces higher drag in comparison with homogeneous conditions while also providing a better agreement with observations. An analysis based on correlations of upstream forest drag with target wind statistics shows that a point above the terrain is most significantly
- influenced by the footprint of a forest area located at about 10 times upstream of its height above ground. When correlations are applied to turbulence, this separation increases five-fold. These findings provide a valuable insight to determine the optimal domain size of a computational domain in forest simulations under neutral atmospheric stratification. Further comparisons of fully uniform vs. limited areas of realistic forest revealed that at heights above 100 m no clear differences in the wind flow are seen. Conversely, comparing flat terrain with the actual topography –with a realistic forest distribution on both cases–
- demonstrated a clear importance of capturing small scale terrain features.

## 1 Introduction

The expansion of wind energy has lead to an increasing interest in the development of projects over remote locations that offer conditions far from the flat and obstacle-free considered as ideal. Forested regions are of interest due to reasons such as the reduced social opposition and the concurrent interests with forestry to share costs for access roads and management. 30 Conversely, they present some of the most challenging wind conditions for the operation of wind turbines: the wind speed is lower, with a stronger vertical shear and higher turbulence intensity compared to winds over terrain with lower vegetation. Indeed, in forested locations wind turbines require more maintenance (Zendehbad et al., 2016). The study of wind flows above forests is far from being restricted to the wind energy community, on the contrary, it is highly relevant in investigations concerning any other structure found on such regions that is subjected to large dynamic loads, such as buildings or bridges, as well as decidedly important in forestry and agricultural applications (Niklas, 1985; Gardiner, 1994; Schindler et al., 2012). 35

Wind flow over forested terrains differs in some aspects compared to that over terrains free of vegetation. The former carries large coherent structures that penetrate the canopy and dominate the turbulence dynamics, including momentum fluxes as well as scalar transport. In some aspects the description of a canopy flow fits more that of a mixing-layer than a boundary layer, an analogy first made by Raupach et al. (1996). This is revealed by the distinctive inflection point in the velocity profile at

- the canopy top as well as other contrasting features to those of a surface layer flow, such as variations in high-order statistical 40 moments, the growth pattern of the turbulence lengthscales or the relations between sweeps and ejections, defined by the directions of components of the shear stress (Gardiner, 1994). The main characteristics of canopy turbulence, gathered from experimental field campaigns and wind tunnel data, were depicted by Raupach et al. (1996) in figures they called a "family portrait", providing a quick reference for the turbulence characteristics for varying canopy heights and densities. These figures
- have since then been reproduced and complemented by other authors, for instance, Brunet (2020). The effect of the forest in this roughness sublayer is conventionally assumed to extend vertically to  $z \approx 2 \sim 3$  times the forest height h. Above it, the wind profile recovers its near logarithmic shape with height z - d, where d is the zero plane displacement height, identified by Thom (1971) and later Jackson (1981) as the level at which the mean drag appears to act on the flow. The asymptotic transition to turbulence statistics similar to those over low vegetation at  $z \approx 2 \sim 3h$  is also supported by measurements over real forests

```
(Arnqvist et al., 2015, 2024).
```

A relevant problem when modelling flows over high roughness at high resolution is that the first grid node above ground ends up embedded deep within the roughness sublayer, where usual flux-gradient expressions are invalid (Basu and Lacser, 2017). Since the roughness sublayer is estimated to be two to three tree heights deep, this problems will be present for microscale simulations of all natural forests. While solutions do exist, they come with unfavourable compromises like setting the height of the first cell undesirably high or moving the stress boundary condition several grid cells up vertically.


The first usage of a second-order closure to model canopy flow was made by Wilson and Shaw (1977) whose 1D formulation includes also a separate source term to account for the forest drag —that has become ubiquitous in CFD studies of canopy flows (see eq. (16)), have been shown able to predict second-order features when comparing with tower measurements (Brunet, 2020). Svensson and Häggkvist (1990) show an early example of employing source terms in the two-equation  $k - \varepsilon$  formulation for

canopy flows (where k is the Turbulence Kinetic Energy or TKE and ε is the turbulence dissipation). As it is inherent to Reynolds-Averaged Navier-Stokes (RANS), a significant challenge is the determination of the modelling coefficients, perhaps even more so in the case of the canopy. This process can involve the usage of other CFD, such as Silva Lopes et al. (2013) or experimental measurements. Among the latter, the model and constants of Sogachev and Panferov (2006) has been favourably used to model wind over heterogeneous forest distributions (Ivanell et al., 2018) and has been later extended (Sogachev, 2009; Sogachev et al., 2012) for the modelling transient atmospheric stability, most suitable for Unsteady RANS (URANS)

calculations as in Sanz Rodrigo et al. (2017, 2021).

In spite of these advances, some of the turbulent flow is characterized by transient coherent eddy structures that are beyond the capabilities of RANS as statistical closure models cannot distinguish between these and incoherent structures (Brunet, 2020). Canopy flows also display distinct features such as sweeps and ejections (downward and upward moving gusts, respec-

- tively), that dominate the turbulence transfer of momentum, heat and mass between the canopy and the atmosphere (Gardiner, 1994; Dupont and Brunet, 2009). Consequently, a faithful representation of the wind dynamics on forested regions requires a modelling technique able to represent the relevant spatial and transient features of these eddy structures. Large-Eddy Simulation (LES) has proven a suitable technique for the reproduction of wind turbulence since larger scales that dominate the flow dynamics are fully resolved, therefore explicitly representing the most significant motions.
- While LES has an extensive use in modelling ABL flows over rough terrains, adjustments to the model equations are required in the case of vegetation canopies. Next to the forest drag acting on the filtered scales, the wakes of leaves and branches precipitate the dissipation of turbulence with respect to the normal breakup of eddies along the energy cascade, an effect sometimes referred to as a "short circuit" of the cascade process (Ayotte et al., 1999) or "spectral short cut" (Finnigan, 2000; Finnigan et al., 2009). This signifies a transfer of energy from the mean flow to the subgrid scales which requires the additional formation of the forest of the forest of the forest of the forest details.

addition of an extra term in the Sub-Grid Scale (SGS) energy budget (Shaw and Patton, 2003).

Since the first LES study of homogeneous canopies by Shaw and Schumann (1992) making usage of an explicit forest drag, multiple studies have continue using this method to model canopy flows. This approach permits, in the first instance, to investigate the characteristics of various statistical moments (Su et al., 1998), carry out one- or two-point correlations and investigate the spectral features of turbulence over forests (Su et al., 2000). LES has shown to be a convenient tool for the

- study of coherent turbulence structures over canopies, e.g. Finnigan et al. (2009), Gavrilov et al. (2011, 2013); Aumond et al. (2013), Bailey and Stoll (2016) and Arnqvist et al. (2024). Investigations about the role of these structures in the processes of turbulent transport within and above canopies are frequently carried out using Quadrant-Hole (QH) analysis technique (Lu and Willmarth, 1973) to identify sweeps and ejections, e.g. Finnigan et al. (2009), Dupont and Brunet (2009), Gavrilov et al. (2011) and Bailey and Stoll (2016). A topic of considerable interest has been the effect of variations in forest density (Dwyer et al., 2011)
- 1997; Dupont and Brunet, 2008a; Adedipe et al., 2020) as well as discontinuities (Silva Lopes et al., 2015; Bou-Zeid et al., 2020) and forest edges (Dupont and Brunet, 2008b, 2009; Boudreault et al., 2017). As pointed out by Bou-Zeid et al. (2020) it has been challenging to develop a clear and coherent theoretical framework that encompasses all of the relevant physics involved in flow over heterogeneous surfaces, particularly when the heterogeneity is less structured.

The advent of Airborne Laser Scans (ALS) has opened new avenues in the field, allowing for an enhanced representation of realistic forests. The point cloud data, consisting of reflections from laser pulses on the ground as well as tree trunks, branches and leaves, are used to create terrain elevation maps as well as to calculate Plant Area Density (PAD) fields and Plant Area Index (PAI) of a desired area (Boudreault et al., 2015; Arnqvist et al., 2020). The utilization of detailed PAD maps in LES provides a significant advantage compared to conventional methods where variations in forest density are represented by modifying an a priori assumed density profile (Dwyer et al., 1997; Dupont and Brunet, 2008a). Even if only information of the forest height is used, the detail in the ALS data has been shown to lead to significantly better wind resource estimation compared to surface descriptions with less detail and accuracy Floors et al. (2018). Examples of studies that have made use of ALS-derived PAD maps are Boudreault et al. (2017); Ivanell et al. (2018), Olivares-Espinosa et al. (2019), Abedi et al. (2021) and Arnqvist et al. (2024), permitting the study of high-order turbulence statistics from realistic forest setups. A particular benefit of using ALS derived PAD fields is the capability to represent the effects of features of the ground and forest heterogeneities along an

- upstream fetch on the wind profiles at a particular location, an attribute referred to as footprint. This aspect has been show in Ivanell et al. (2018) and Arnqvist et al. (2019) where the usage of PAD fields in LES enables to reproduce properties in the wind profile hypothesized to stem from characteristics in the footprint of different incoming wind directions. The question of how large such an upstream region needs to be is partly the subject of the present work.
- An additional benefit of using PAD fields in numerical simulations is that variations in tree height as well as clearings are directly incorporated and their subsequent effect in the canopy shear stress is naturally assimilated. Indeed, as shown by Silva Lopes et al. (2015) and Janzon et al. (2023), landscapes of alternating forests and clearings yield a shear stress that in average is larger than the sum of the equilibrium stresses over homogeneous patches, resulting in a higher effective  $z_0$ . On the contrary, Boudreault et al. (2017) found using RANS that the horizontal heterogeneities induce higher turbulence predominately at the canopy top while decreasing the displacement height that in turn leads to higher velocities compared to a
- homogeneous canopy. While the literature thus is conflicting on the impact of heterogeneous forest cover on the wind above, it is clear that to represent a real forest with constant tree height and homogeneous density profile is a crude approximation.

For wind power deployment in forested landscapes, the main focus is on shear and turbulence magnitude, with directional shear, integral lengthscales and other turbulence statistics also being of interest (Robertson et al., 2019). As it is still an open question what methodology is suitable to predict such properties in the wind given a specific site, this work scrutinize the use


o of realistic PAD profiles in LES and investigates to which extent its use can improve predictions of wind statistics owing to the character of the upstream forest footprint.

The layout of this work is as follows: First, the requirements to reach statistically significant conclusions are examined, followed by a description of the field measurements, the flow model and the post-processing of the data. This is followed by the results, divided in 3 parts: first with focus on the verification of the PAD approach, then the validation against field


measurements and finally with respect to the impact of the upstream forest cover. The paper finishes with conclusions of the most important findings for each of these parts as well as recommendations and reflections regarding future research on the topic.

#### 2 Assessment of requirements to test site specific CFD capability

The following section presents a brief outline of the statistical requirements for numerical simulations that permit to establish
whether differences in results arise due to distinct upstream conditions rather than stochastic variability. The aim of this is to confirm that the modelling technique is able to reproduce differences owed to surface heterogeneities.

### 2.1 Requirements on the length of the LES run

The inclusion of the physical mechanisms into an LES model required to reproduce its response to the boundary conditions representing a particular site (PAD and topography in the present case) does not guarantee by itself the capability of reproduc-

ing the wind flow with a reasonable precision. Additional requirements are necessary regarding the accuracy of the validation measurements, boundary conditions and wind statistics. Even under the assumption of an accurate measurement of the parameters employed to define the boundary conditions as well as wind statistics (or within a negligible error), the statistical uncertainty of the LES simulation itself must be such that any random error is smaller than the site specific response in the wind statistics. According to Lumley and Panofsky (1964) the statistical uncertainty for wind speed can be estimated by

$$\frac{\sigma_{\overline{u}}}{\overline{u}} = \left(\frac{2\mathcal{T}_1 \sigma_u^2}{T\overline{u}^2}\right)^{1/2},\tag{1}$$

where  $\overline{u}$  is the mean wind,  $\sigma_{\overline{u}}$  is the random error of the mean wind,  $\mathcal{T}_1$  is the integral time scale of u,  $\sigma_u^2$  is the variance of the instantaneous streamwise wind and T is the length of the simulated time series. An estimation of the magnitude of the relative random error can be made by assuming that  $\mathcal{T}_1 \sim 10z/\overline{u}$ , where z is the height above ground, and  $\sigma_u^2/U^2 \sim 10^{-2}$ . Assuming  $z/\overline{u} \sim 10$  implies that it would be necessary to simulate at least 20000 s to get below a relative random error of 1%.


The relative random error for second order moments can, according to Lenschow et al. (1994), be estimated by

$$\frac{\sigma_{\overline{u_i}\overline{u_j}}}{\overline{u_i'u_j'}} = \left(\frac{2\mathcal{T}_{ij}}{T}\right)^{1/2},\tag{2}$$

where  $T_{ij}$  is the integral time scale of the second order moment. Assuming again that  $T_{ij} \sim 10z/\overline{u}$  implies that the relative random error for  $\overline{u'_i u'_j}$  is 10 % at 100 m height for a simulation of length 20000 s. To get to a relative random error below 1 % would require to simulate  $2 \times 10^6$  s, or more than 23 days of physical time!


Using scaling arguments, a relative difference in mean wind between two different wind directions at a single site due to different surface roughness can be estimated. For wind speed we have that

$$\frac{\overline{u}}{u_*} = \frac{1}{\kappa} \ln z / z_0,\tag{3}$$

where  $u_*$  is the friction velocity,  $\kappa$  is the von Kármán constant and  $z_0$  is the roughness length. Taking the difference between two directions (direction 1 and 2) we get

$$\Delta \overline{u} = u_{*_1} \frac{1}{\kappa} \ln z / z_{0_1} - u_{*_2} \frac{1}{\kappa} \ln z / z_{0_2}.$$
 (4)

Furthermore, if the difference in roughness lengths between the two directions is relatively small, the ratio of roughness lengths between the two directions is much larger than the ratio in wind speed or friction velocity so that the latter two ratios can be approximated to 1. This allows to extract  $\overline{u}$  from the right hand side:

$$\frac{\Delta \overline{u}}{\overline{u}} \approx \frac{\ln z_{0_2}/z_{0_1}}{\ln z/\overline{z_0}},\tag{5}$$

wh

where  $\overline{z_0}$  is the mean roughness length of the two directions. To provide an example, if one direction has a fetch with a roughness length of 2 m and another direction has a roughness length of 1.5 m, the relative wind speed difference between them at 100 m height would be approximately 7 % according to eq. (5).

Using the logarithmic law, eq. (3), we can also estimate the relative difference in shear stress owing to a (small) difference in  $z_0$ :

$$\Delta u_*^2 = \left(\frac{\kappa \overline{u_1}}{\ln z/z_{0_1}}\right)^2 - \left(\frac{\kappa \overline{u_2}}{\ln z/z_{0_2}}\right)^2.$$
 (6)

Assuming that we want to estimate the difference in  $u_*^2$  for a fixed wind speed at height z, and assuming we can approximate the mean shear stress as  $[\kappa \overline{u}/\ln(z/\overline{z_0})]^2$  we get the following expression for the relative difference in shear stress,

$$\frac{\Delta u_*^2}{\overline{u_*^2}} \approx \ln(z/\overline{z_0})^2 \left(\frac{1}{\ln(z/z_{0_1})^2} - \frac{1}{\ln(z/z_{0_2})^2}\right). \tag{7}$$

This expression indicates that at 100 m height, we can expect a relative difference in the shear stress of 14 % between a 170 direction with  $z_0 = 2$  m compared to a direction with  $z_0 = 1.5$  m.

From the above estimations we can conclude that it is reasonable to expect of a LES model that has simulated 20000 s physical time to reproduce differences in mean wind speeds between two different wind directions at a particular site, but that validating the models' ability to detect differences in higher order statistics would be on the limit of statistical uncertainty.

## 2.2 Requirements on the length of the model domain

In order to roughly estimate the requirements on domain size to capture the footprint of a heterogeneous terrain we apply the following scaling arguments: if there is a heterogeneity in the wind field owing to a heterogeneity in the surface roughness or topography, its local impact on the streamwise wind can be estimated by the size of the streamwise advection term in the momentum equation,

$$\overline{u}\frac{\partial\overline{u}}{\partial x} \sim \overline{u}\frac{\Delta\overline{u}}{x}.$$
(8)

On the other hand, the most important term that tends to even out heterogeneity in the wind field is the vertical shear stress divergence,

$$\frac{\partial \overline{u'w'}}{\partial z} \sim \frac{u_*^2}{z}.$$
(9)

Assuming both  $\Delta \overline{u}$  and  $u_*$  are of the order  $0.1\overline{u}$  we can estimate the ratio between the strength of the advection term to the shear stress divergence as

$$\frac{\overline{u}z}{u_*x} \sim \frac{10z}{x}.\tag{10}$$

For upstream distances shorter than 10z the advection will dominate and the wind field will be characterized the upstream heterogeneity. On the other hand, if advection is to be completely negligible, either the upstream surface conditions must be homogeneous or the distance x to the heterogeneity must satisfy  $x \gg 10z$ . To quantify, we expect that an upstream heterogeneity lying further away than 100z would contribute with less than 10% of the momentum balance of the flow. Thus, we conclude that to capture most of the effects from an upstream surface heterogeneity, the model domain should extend

### 3 Site description and measurements

downstream roughly 100 times the highest height of the wind turbine rotor.

### 3.1 Site description and surface data

Metmast measurements correspond to an experimental campaign at the location of Ryningsnäs, a forested and mildly complex region in the southern part of Sweden, at about 30 km from the coast of the Baltic sea.

While part of the simulation cases in this work assume idealized forest conditions, LES are also produced to represent on-site conditions whose results are compared to measurements. Following Ivanell et al. (2018); Arnqvist et al. (2019), three different wind directions were modelled in order to see if the impact of the upstream vegetation cover is the same in the LES as in the observations. The different cases are summarized in Table 3. While all cases have predominately forest cover upstream for

at least 30 km, surface characteristics vary for the three incoming wind directions: case R1 is characterized by a 400 m wide clearing just upstream of the met tower, case R2 has a valley covered with low vegetation (crops) dominating the fetch between 5 and 10 km upstream while case R3 is impacted by the same valley, but to much lesser degree and as such has less distinct features in its fetch.

The characterization of the surface data was made by analyzing point clouds from ALS with the method of Arnqvist et al. (2020). The point cloud was used to compute PAD, ground height and vegetation height in a 10 m ×10 m grid. The vertical resolution of the PAD data was 1 m. Detailed descriptions of the vegetation cover in the three directions are given in Ivanell et al. (2018).

#### 3.2 Measurements

The measurements were taken from a 140 m high met tower operated between 2009 and 2012. The instrumentation consisted of 6 Metek USA-1 3D-sonics and 7 Thies first class cup anemometers. The measurement heights were 40, 59, 80, 98, 120, and 137.7 m for the sonics and 25.5, 40.1, 60.5, 80.1, 95.85, 120.75, and 137.6 m for the cups. More details of the measurements can be found in Arnqvist et al. (2015) and Bergström et al. (2013).

## 3.2.1 Statistical processing

For wind speed the average between the cup anemometers and the sonic anemometers was used. The shear exponent of the power law for the wind speed was calculated between two height levels in the tower as