# Peer review of "Modelling of wind flows over realistic forests with LES"

_Wind Energy Science, 2025_

## Referee Comment (RC2)

Review of the paper "Modelling of wind flows over realistic forests with LES".

Submitted to Wind Energy Science

Article number #: wes-2025-114

October 12, 2025

Recommendation: Minor revision.

**Summary:**

This paper presents a large-eddy simulation framework for modeling wind flow over realistic forested terrain using airborne laser scan data to derive detailed plant area density and topography. The topic is of considerable interest, and the methodology is robust. This paper makes a contributions to the field of wind energy by quantifying how terrain and vegetation heterogeneity influence wind flow and turbulence characteristics relevant to wind resource assessment. However, several issues must be addressed before the manuscript can be recommended for publication. My comments are categorized as either 'Major concerns' or 'Minor concerns', with the former focusing on conceptual technical critiques, and the latter highlighting grammatical and spelling errors.

**Major concerns:**

- (1): Please emphasize more clearly in the Introduction how this work advances beyond earlier studies such as Ivanell et al. (2018), Arnqvist et al. (2019, 2024), or Boudreault et al. (2017). A short paragraph summarizing specific new insights would strengthen the positioning.
- (2): The discussion in Section 2.1 provides useful scaling arguments, but the justification for the selected simulation length (20,000 s) would benefit from including convergence diagnostics, such as the time evolution of the mean and variance at representative heights. It would also be helpful to express the simulation duration in terms of large-eddy turnover times, which would facilitate comparison and reproducibility by other researchers.
- (3): The formulation of the additional subgrid dissipation term ("spectral short-cut") is clearly described. It would be helpful to show its quantitative impact—for example, a comparison of vertical profiles with and without the term for one case (perhaps as supplementary material).
- (4): The fixed drag coefficient Cd=0.2 may not capture species or height variability; a short justification or sensitivity test could be valuable.
- (5): The footprint analysis (Section 6.3) provides interesting insights regarding upstream influence. This is a strong result—consider relating it to prior footprint models to highlight consistency or differences.

**Minor concerns:**

• (1): Ensure consistency in the reference list—some entries include DOIs while others do not. Please include DOIs for all references where available.

**References**

---

## Author Comment (AC1)

**Reply to comments by Anonymous Referee #2**

Manuscript wes-2025-114 by Olivares-Espinosa, H. and Arnqvist, J.

We would like to thank the reviewers for the time they have invested reviewing this work as well for the insight they provide. Their remarks and suggestions have contributed meaningfully to the improvement of this work.

We begin by noting that we have made a change to the title to add the word *footprint* which we believe better reflects the scope of the investigation. A large part of the manuscript revolves around this topic and we wish to display this from the title, something that was missing in the first submission. We also think that this change reflects some of your comments about emphasizing this study.

Below we address the comments one by one. Reviewer's comments and questions are in slanted font and each point is appended by the letter **R** in boldface with an answer appended by **A.R** in boldface red.

**Recommendation***: Minor revision.* **Summary:** *This paper presents a large-eddy simulation framework for modeling wind flow over realistic forested terrain using airborne laser scan data to derive detailed plant area density and topography. The topic is of considerable interest, and the methodology is robust. This paper makes a contributions to the field of wind energy by quantifying how terrain and vegetation heterogeneity influence wind flow and turbulence characteristics relevant to wind resource assessment. However, several issues must be addressed before the manuscript can be recommended for publication. My comments are categorized as either 'Major concerns' or 'Minor concerns', with the former focusing on conceptual technical critiques, and the latter highlighting grammatical and spelling errors.*

**Major concerns:**

**R2-1** *Please emphasize more clearly in the Introduction how this work advances beyond earlier studies such as Ivanell et al. (2018), Arnqvist et al. (2019, 2024), or Boudreault et al. (2017). A short paragraph summarizing specific new insights would strengthen the positioning.*

**A.R2-1** Thank you for pointing this out. Below is a list of what we consider being the most important contributions of the study. We added a paragraph near the end of the introduction to reflect this.

(a) Comprehensive model verification compared to above listed references

(b) More detailed validation against observations than above listed references

(c) Quantified footprint study (lacking from above mentioned references)

    – Complementary method to estimate footprint to existing methods

    – The relative importance of terrain and heterogeneity

      – Comprehensive recommendations for modelling choices based on the above points as well as the scaling arguments in Section 2.1

**R2-2** *The discussion in Section 2.1 provides useful scaling arguments, but the justification for the selected simulation length (20,000 s) would benefit from including convergence diagnostics, such as the time evolution of the mean and variance at representative heights. It would also be helpful to express the simulation duration in terms of large-eddy turnover times, which would facilitate comparison and reproducibility by other researchers.*

**A.R2-2** In principal we agree with the reviewer on this suggestion. However, as stated in Sec. 4.2.1, simulations were started with initial conditions from a previous simulation to speed up convergence. This means that the rate of convergence is somewhat dependent on the specifics of this initial field and would be of limited use for others repeating the study. After initial tests were the rate of convergence was observed, the spinup period was not saved in order to save storage space, subsequently we do not have the spinup for all of the cases. Nevertheless we are happy to provide the plots for case F2 where the spinup was saved.

Fig. 4 below displays the evolution of the components of velocity and their variances during the $120 \times 10^3$ s of the simulation run previous to the sampling period, i.e. the spinup. Next to the evolution of the said values, the curves also add a cumulative average of each quantity, to demonstrate that after some period of time, the mean value has converged and more data will not alter the average. For the heights shown in the figure, this occurs sometime after 15 h of simulation. For the other cases, even if convergence took longer to be achieved, it is highly unlikely that it would be reached after the spinup period employed.

Finally, we would like to clarify that the purpose of the analysis in Section 2.1 is not to discuss the convergence of the LES in the canonical sense. Instead, we focus on the requirement to converge enough to facilitate detection of small differences in the wind statistics arising from slightly different upstream forest cover. This is a stricter requirement than what one typically used. This, together with the point above and the fact that the paper is already on the long side has contributed to our decision of leaving the plots out of the manuscript.

For the most relevant cases discussed in the Verification Section 6.1, F1, F6 and F9, the evolution of the means of velocities and variances during the acquisition period can be seen in Figures 5, 6 and 7. In those, it can be seen that the cumulative means stabilize fairly quickly, but as shown with the case F2, this sampling period follows a flow statistically converged. The values of the spinup and sampling periods in eddy-turnover times have been added to the text in Sec. 4.2.1.

**R2-3** *The formulation of the additional subgrid dissipation term ("spectral short-cut") is clearly described. It would be helpful to show its quantitative impact—for example, a comparison of vertical profiles with and without the term for one case (perhaps as supplementary material).*

**A.R2-3** We believe that what the reviewer suggests is displayed in Figure 5. As per Table 2, the cases F4 and F5 lack the enhanced dissipation, but it is included in F3. As you can see in Figure 5 (c) and (d), this has implications for the TKE close to the

[Figure]

**Figure 4.** Evolution of velocity components (left column) and variances (right column) for case F2 at different heights for spinup period. The high-frequency value, resulting from the ensemble average of the 9-column data, is shown with low opacity whereas the solid lines correspond to their cumulative average.

[Figure]

**Figure 5.** Evolution of velocity components (left column) and variances (right column) for case F1 at different heights during the sampling period. The high-frequency value, resulting from the ensemble average of the 9-column data, is shown with low opacity whereas the solid lines correspond to their cumulative average.

[Figure]

**Figure 6.** Evolution of velocity components (left column) and variances (right column) for case F6 at different heights during the sampling period. The high-frequency value, resulting from the ensemble average of the 9-column data, is shown with low opacity whereas the solid lines correspond to their cumulative average.

[Figure]

**Figure 7.** Evolution of velocity components (left column) and variances (right column) for case F9 at different heights during the sampling period. The high-frequency value, resulting from the ensemble average of the 9-column data, is shown with low opacity whereas the solid lines correspond to their cumulative average.

canopy, a relatively smaller impact on wind-energy relevant heights (b) while (a) shows that for wind speed the effect is very small, hardly noticeable at all.

**R2-4** *The fixed drag coefficient Cd=0.2 may not capture species or height variability; a short justification or sensitivity test could be valuable.*

**A.R2-4** Thank you for the suggestion. We acknowledge that the value of the drag coefficient is probably the most significant remaining weakness in the modelling strategy of flows through canopies. We have updated the introduction and discussion accordingly. We don't have a strong justification of the constant value used in the study, other than that there is, to our understanding, a lack of suitable, well founded, parametrizations for the variation of $C_D$. Our suggestions for future work reflect this opinion. Since we don't have any good testable hypothesis for the variation of $C_D$, we believe there would be relatively little contribution from a sensitivity study. The results of which would likely not be general anyway. We are aware that candidates for the cause of variation of $C_D$ naturally exist: Reynolds number sensitivity, streamlining effects, effects of atmospheric stratification, resolution issues for both PAD and the flow, etc. but since there are still so many uncertainties we maintain that the problem is best tackled experimentally and/or with dedicated high resolution simulations at this stage.

**R2-5** *The footprint analysis (Section 6.3) provides interesting insights regarding upstream influence. This is a strong result—consider relating it to prior footprint models to highlight consistency or differences.*

**A.R2-5** Thank you for the suggestion. We have added references to previous work on footprint and blending height (mainly in the introduction) and highlighted some complementary information brought by this study in Section 6.3 and 6.4.3. We have also added a discussion on the impact of very large scale motions to the estimation of the footprint.

**Minor concerns:**

**R2-6** *Ensure consistency in the reference list—some entries include DOIs while others do not. Please include DOIs for all references where available.*

**A.R2-6** DOIs have been added to all reference entries. In the case where a DOI could not be found, URL have been used instead.

---

## Author Comment (AC2)

**Reply to comments by Anonymous Referee #1**

Manuscript wes-2025-114 by Olivares-Espinosa, H. and Arnqvist, J.

We would like to thank the reviewers for the time they have invested reviewing this work as well for the insight they provide. Their remarks and suggestions have contributed meaningfully to the improvement of this work.

We begin by noting that we have made a change to the title to add the word *footprint* which we believe better reflects the scope of the investigation. A large part of the manuscript revolves around this topic and we wish to display this from the title, something that was missing in the first submission. We also think that this change reflects some of your comments about emphasizing this study.

Below we address the comments one by one. Reviewer's comments and questions are in slanted font and each point is appended by the letter **R** in boldface with an answer appended by **A.R** in boldface red. A bibliography section of citations made exclusively for the reviewers' replies follows.

**R1-1** *Whether or not the forest is modeled explicitly, if the imposed roughness length z0 in the wall model is truly representative of a forest canopy, the resulting profiles well above the canopy should converge. The fact that they differ indicates that the specified z0 is not forest-equivalent. Figures 8 and 9 should be rerun with a truly forest-equivalent z0.*

**A.R1-1** We agree with this observation, the $z_0 = 0.65$ in case F9 is not an equivalent roughness value to the PAI/PAD employed in F1. As mentioned in the manuscript, the "equivalent" $z_0$ and $d$ values were estimated following the principles found in Mohr et al. (2018). A better estimate could be made by applying roughness-sublayer corrections for the flux-gradient relationship, but doing so would still not solve the issue, as the magnitude of the roughness length would introduce limitations in the grid resolution of the LES. For $h_f = 20$ m, Arnqvist et al. (2015) reported a value of $z_0 = 3$ m and $d = 13$ m for neutral conditions, obtained by fitting wind velocity profiles to the logarithmic Monin-Obukhov relationship. The usage of roughness lengths of this magnitude is an important practical problem since standard flux-gradient relationships hold only within the inertial sublayer. Indeed, the suggestion of Basu and Lacser (2017) to maintain the applicability

of the Monin-Obukhov similarity theory is to set the first cell above the ground $z_1 > 50z_0$, a suggestion that comprises the simulation of stable ABLs. For neutral boundary layers –albeit not atmospheric–, the collection presented by Huang et al. (2016) of DNS and wind tunnel experiments translate into a threshold of $z_1 > 22z_0$ after applying the same principles Basu and Lacser (2017) –essentially the relation $h \approx 10z_0$ from Townsend (1976)–. To set $z_1$ at such heights is not desirable for wind energy applications.

To correct the misleading "equivalent roughness", we have added the description of how the $z_0$ and $d$ are obtained. It is now mentioned that these are calculated based on parametrizations that aim at providing tuning-free estimations based on PAI and tree height, referred to as "PAI-derived". All previous instances of "PAD-equivalent" have been changed to fit this terminology, including Table 2.

On an additional note, wee acknowledge that a suitable combination of $z_0$ and $d$ together with roughness sublayer correction would exist, especially in an analytical sense, but that using such a setup has the drawback of increased subjectivity because of the empirical nature of the roughness sublayer correction. We have added to the discussion on subjectivity in Page 3 of the introduction and Section 6.1 and Section 6.4.1.

**R1-2** *At the Reynolds numbers considered, one would expect to resolve the -1 spectral scaling rather than the classical -5/3 scaling. The reference scaling should be adjusted, and additional discussion of the -1 scaling is needed.*

**A.R1-2** Thank you for pointing that out. We agree with the reviewer that parts of the spectrum should likely be influenced by -1 scaling and we have subsequently added the -1-exponent reference to Figure 6. We have referred to Katul et al. (2012) on the theory on -1. We have limited the -1 reference to the LES inter-comparison section, with the idea that the -1 scaling will be less evident in less idealized simulations (and certainly in the measurements) where varying atmospheric stratification and meso- as well as synoptic-flow modulations add variance at low frequencies.

**R1-3** *The present method of computing SGS TKE assumes isotropy of the unresolved turbulence. This assumption is unlikely to be valid for canopy-driven flows. The discussion on SGS TKE could be removed, or at least significantly qualified.*

**A.R1-3** This is valuable observation and we thank you for noting it. Certainly, the anisotropy caused by the plant elements does also occur at the SGS scales and its inaccurate representation into the model can affect the of production, dissipation and transport of subgrid TKE. The use of eddy-viscosity modelling that relates the SGS stress tensor to the resolved rate of strain and the subgrid viscosity entails the assumption that the dissipation rate is the same in all directions. In directionally dependent environments such as canopy flows, the subgrid dissipation could be underestimated in some directions while overestimated in others (this could include backscatter but it is not considered in our subgrid models). Another issue could be the damping of coherence structures relating to canopy phenomena like sweeps and ejections. The usage of a TKE transport equation as the one employed in our work (Yoshizawa and Horiuti, 1985; Yoshizawa, 1986) can help to alleviate some of these issues as it can capture —in principle— the changes in SGS magnitude across different spatial directions, especially in comparison with Smagorinsky modelling. Yet, it cannot reproduce the missing anisotropy as it is based on the same modelling of the SGS stress tensor.

Naturally, these issues would become more important for coarse resolutions as the proportion of bulk energy in the subgrid range increases. If the resolution is sufficient, the omission of anisotropy in the assumptions of subgrid model should not have a major effect in the energy transfer. Inagaki and Kobayashi (2023) study the effects of considering the anisotropic part of the SGS stress in channel flows and report that coarse (sharp-cut Fourier) filters have an effect on the reproduction of spanwise velocity fluctuations and the generation of coherent structures. Moreover, Bhuiyan and Alam (2020) show that including the effects of vortex stretching and coherent structure magnitude in a dynamic SGS model leads to a noticeably increase in subgrid TKE in comparison with SGS models with a TKE transport equation (note this work remains unpublished and it is available only as a preprint).

Yet, it should be noted that wind statistics acquired from forested areas reveal that variances are sharply reduced inside the forest and around the canopy top in comparison with the flow above. An example of this can be found in the work of Arnqvist (2013, Fig. 4.3 in page 26) where the variances not only become much smaller towards the interior of the canopy but also they become similar in magnitude, in contrast to their anisotropic characteristic above.

A discussion to the effect of the above as been included in the description of the model in Sec. 4.1 and in the discussion of the SGS modelling in Sec. 6.1.1.

**R1-4** *At times, the paper reads more like a comprehensive technical report than a sharply focused scientific article. A sharper emphasis on novelty would strengthen the contribution.*

**A.R1-4** We acknowledge that the content of the manuscript is extensive and above the average number of pages. There is a substantial amount of material that perhaps could have been split into two separate submissions, for example, one focusing on the verification of the model and another on the characteristics of the flow over realistic forests, including its footprint. In contrast, presenting all this work in a single submission carries the intention of strengthen the confidence in the results obtained with the methodology, in particular those regarding the observations about the usage of ALS-derived PAD fields, domain size and forest footprint.

We have made our best effort in highlighting the most relevant results of our investigation. This is why we have included separate subsections in the conclusions for *Verification, Validation* and *Capturing the footprint of the forest* which also carries the intention of emphasizing the novelty of the study. We kindly ask the reviewer to let us know if she/he believes that further reorganization of the text is necessary to better highlight the novelty of our results.

**R1-5** *The long lifetime of streamwise-elongated streaks in turbulent boundary layers is well established. Consequently, the footprint of upstream terrain should not be expected to scale with forest height, but rather with O(10) times the boundary-layer height. The authors should acknowledge this existing work and place their findings in that context.*

**A.R1-5** Thank you for pointing out this oversight in the first manuscript. We agree that long streaks and/or Very-Large-Scale Motions (VLSMs) should contribute with an outer length scale that in effect extends the footprint. We have added relevant references to the introduction which we come back to in the discussion. In addition, we now also acknowledge this effect on the spectra Section 6.1.2. We want to clarify though that we still consider the main contribution to the

footprint will be from surface related scales and that this seems to be supported by literature, see for instance Paleri et al. (2022). While streaks are clearly visible in the simulations, their impact on the footprint would be limited to large scales since the ratio $u/\sigma_w$ varies considerably less than $u$ or $\sigma_w$ individually. Finally, some of the streaks do seem to be connected to specific upstream surface features and would then be partly different from the streaks discussed in the literature of VLSM.

**References in reply to reviewer's comments**

Bhuiyan, M. A. S. and Alam, J. M.: Subgrid-scale energy transfer and associated coherent structures in turbulent flow over a forest-like canopy, arXiv preprint arXiv:2010.01463, https://doi.org/10.48550/arXiv.2010.01463, 2020.

Huang, G., Simoëns, S., Vinkovic, I., Le Ribault, C., Dupont, S., and Bergametti, G.: Law-of-the-wall in a boundary-layer over regularly distributed roughness elements, Journal of Turbulence, 17, 518–541, 2016.

Inagaki, K. and Kobayashi, H.: Analysis of anisotropic subgrid-scale stress for coarse large-eddy simulation, Physical Review Fluids, 8, 104 603, https://doi.org/10.1103/PhysRevFluids.8.104603, 2023.

Townsend, A.: The structure of turbulent shear flow, Cambridge university press, 1976.